# Novel Insights into the Mechanism Underlying High Polysaccharide Yield in Submerged Culture of *Ganoderma lucidum* Revealed by Transcriptome and Proteome Analyses

**DOI:** 10.3390/microorganisms11030772

**Published:** 2023-03-17

**Authors:** Qiong Wang, Mengmeng Xu, Liting Zhao, Lei Chen, Zhongyang Ding

**Affiliations:** 1Key Laboratory of Industrial Biotechnology, Ministry of Education, School of Biotechnology, Jiangnan University, Wuxi 214122, China; 2Key Laboratory of Carbohydrate Chemistry and Biotechnology, Ministry of Education, School of Biotechnology, Jiangnan University, Wuxi 214122, China; 3National Engineering Laboratory for Cereal Fermentation Technology, Jiangnan University, Wuxi 214122, China

**Keywords:** polysaccharide, biosynthesis, transcriptome, proteome, *Ganoderma lucidum*

## Abstract

Polysaccharides are crucial dietary supplements and traditional pharmacological components of *Ganoderma lucidum*; however, the mechanisms responsible for high polysaccharide yields in *G. lucidum* remain unclear. Therefore, we investigated the mechanisms underlying the high yield of polysaccharides in submerged cultures of *G. lucidum* using transcriptomic and proteomic analyses. Several glycoside hydrolase (GH) genes and proteins, which are associated with the degradation of fungal cell walls, were significantly upregulated under high polysaccharide yield conditions. They mainly belonged to the GH3, GH5, GH16, GH17, GH18, GH55, GH79, GH128, GH152, and GH154 families. Additionally, the results suggested that the cell wall polysaccharide could be degraded by GHs, which is beneficial for extracting more intracellular polysaccharides from cultured mycelia. Furthermore, some of the degraded polysaccharides were released into the culture broth, which is beneficial for obtaining more extracellular polysaccharides. Our findings provide new insights into the mechanisms underlying the roles that GH family genes play to regulate high polysaccharide yields in *G. lucidum*.

## 1. Introduction

The fungus *Ganoderma lucidum* (Lingzhi) is a well-known dietary supplement and traditional herbal medicine with various pharmacological properties [1]. Polysaccharides, including fruiting body polysaccharides, intracellular polysaccharides (IPS) from cultured mycelium (also known as mycelial polysaccharides), and extracellular polysaccharides (EPS) from cultured broth, are recognized as the major bioactive molecules of *G. lucidum* [2,3]. Various *G. lucidum* polysaccharides with different structures have been isolated and elucidated for their antioxidant, immunostimulating, antitumor, and antimicrobial properties [4,5,6]. Nonetheless, the integrated synthesis pathways and mechanisms underlying the high yield of polysaccharides in *G. lucidum* have not been reported. Moreover, the roles of polysaccharide synthesis-related enzymes in *G. lucidum* must be comprehensively elucidated.

Generally, the integrated biosynthetic pathway of polysaccharides includes the following steps: the synthesis of NDP-sugar precursors, assembly through glycosyltransferases (GTs), and polymerization and export of EPS [7,8]. In *G. lucidum*, the nucleoside sugar biosynthetic pathway and the roles of many synthesis-related enzymes have been elucidated [9,10]. Particularly, phosphoglucose isomerase (GPI), α-phosphoglucomutase (PGM), and UDP-glucose pyrophosphorylase (UGP) enzymes involved in nucleoside sugar biosynthesis in *G. lucidum* are the most widely studied. Furthermore, the activities of these three enzymes are correlated with the production of IPS and EPS in *G. lucidum* [11,12,13,14,15]. The production of IPS and EPS is increased by the overexpression of PGM, UGP, and phosphomannomutase (PMM) genes, respectively [7,16,17]. Moreover, strains with silenced PGM and UGP genes exhibit a reduced polysaccharide production [18,19].

GTs are a highly diverse group of enzymes with minimal homology, even among enzymes that share the same substrate specificity [20,21]. GTs facilitate polysaccharide assembly. Enzymes catalyze the transfer of sugar moieties from activated donor molecules to specific acceptor molecules to form oligosaccharides, polysaccharides, and other glycoside compounds. The GT reportedly involved in polysaccharide biosynthesis in *G. lucidum* is 1,3-β-glucan synthase (GLS), which belongs to the GT 48 family. The transcription levels of *GLS* have been demonstrated to be upregulated in *G. lucidum*-engineered strains producing high yields of EPS and IPS [7,22]. In *Grifola frondosa*, silencing transformants of GLS showed a decreased mycelial biomass and exo-polysaccharide production [23,24]. After the assembly process, these macromolecules must be exported from the cell membranes. However, the mechanism has not been observed in mushrooms. 

The polysaccharide biosynthetic pathway is complex, and few studies have focused on the overall pathway in mushrooms. In other fungi, especially pathogenic fungi, the synthesis of biofilm polysaccharides is the most studied. Galactosaminogalactan (GAG), galactomannan, (1→3)-α-glucan, and (1→3)-β-glucan are the main biofilm polysaccharides, and their biosynthetic pathways have been reported [25,26]. Biofilm polysaccharides are under the control of the glycoside hydrolase (GH) family of enzymes. However, the mechanisms underlying this enzymatic process have not yet been elucidated in *G. lucidim*. In addition, biofilm polysaccharides are similar in composition to polysaccharides of the fungal cell wall [27,28,29]. The cell wall polysaccharide biosynthetic pathway also contains cell wall-associated glycosyl hydrolases, which are responsible for remodeling de novo synthesized polysaccharides and establishing the three-dimensional structure of the cell wall through hydrolase and transglycosidase activities [29,30]. 

Transcriptomes and proteomes have been widely used to study specific biological processes in *G. lucidum* as powerful tools for analyzing differentially expressed genes (DEGs) and differentially expressed proteins (DEPs) [31,32,33]. Therefore, in this study, we have particularly investigated the key GT and GH family genes and proteins associated with high polysaccharide yields in *G. lucidum* through transcriptome and proteome analyses. Furthermore, we used glucose and xylose as a carbon source because the carbon source is a vital factor in cell growth and polysaccharide production. This work provides important insights into the mechanisms that govern a high polysaccharide yield in *G. lucidum*.

## 2. Materials and Methods

### 2.1. G. lucidum Strain and Culture Conditions

The strain *G. lucidum* CGMCC5.26 was purchased from China General Microbiological Culture Collection Center and maintained on potato dextrose agar slants at 4 °C. The seed medium and preculture details of *G. lucidum* have been described previously [9,11]. Briefly, the medium was composed of glucose (20 g/L), yeast nitrogen base without amino acids (5 g/L), tryptone (5 g/L), KH_2_PO_4_ (4.5 g/L), and MgSO_4_·7H_2_O (2 g/L) at initial pH. The seed was in a 250 mL flask containing 80 mL of medium and was kept at 30 °C on a rotary shaker (150 rpm). In the fermentation medium, the carbon sources were glucose and xylose, and the other components were the same as those in the seed medium. Before inoculating, the seeds were homogenized using an IKA T10 basic homogenizer (IKA, Königswinter, Germany). Briefly, the sterilized dispersing elements of homogenizer inserted the seed medium into a 250 mL flask, and homogenized with the fifth level for 1 min. Then, the homogenized seed medium was centrifuged at 10,000× *g* for 5 min, and 3.5 g of wet weight cells/L were inoculated into a 500 mL flask with 150 mL of medium. The submerged culture was grown on a rotary shaker at 150 rpm and 30 °C for eight days.

### 2.2. Sampling and Analysis of Biomass and Polysaccharides

Mycelia were harvested by centrifugation (10,000 rpm) for 10 min. The precipitate was washed thrice with distilled water and dried by lyophilization. Biomass was determined using the gravimetric method of determining the dry cell weight. For obtaining IPS, dry mycelia extracts were prepared by boiling the mycelia for 3 h. EPS were obtained from the harvested fermentation broth. Mycelia extract and fermentation broth were centrifuged at 10,000 rpm for 10 min, and the supernatants were collected. Subsequently, crude polysaccharides were precipitated by adding four volumes of 95% (*v*/*v*) ethanol and they were left overnight at 4 °C. The precipitate was then separated by centrifugation at 10,000 rpm for 10 min and washed twice with 75% (*v*/*v*) ethanol. The precipitate was dried at room temperature to remove residual ethanol. The polysaccharide content was assayed using the phenol–sulfuric acid method [34]. Briefly, the dried polysaccharide was dissolved in distilled water. Two milliliters of polysaccharide solution was pipetted into a tube, and 1 mL of 6% phenol was added. Then, 5 mL of sulfuric acid was added to the tube, which was then shaken rapidly for better mixing. After cooling to room temperature, the absorbance of the sample was measured at 490 nm. 

### 2.3. Transcriptome and Proteome Analyses

A total of 18 cDNA libraries at three time points, namely on days 4, 6, and 8, were prepared for transcriptome analysis. RNA isolation, cDNA library construction, sequencing library preparation, and high-throughput sequencing were performed by Novogene Bioinformatics Technology Co. Ltd. (Beijing, China; http://www.novogene.com, accessed on 3 September 2020). This procedure was performed as previously reported [35].

After culturing for eight days, mycelia were collected for proteome analysis using label-free methods. Total protein extraction, protein quality testing, trypsin treatment, and LC-MS/MS analysis were also conducted by Novogene Bioinformatics Technology Co., Ltd. The details are summarized in the Appendix A.

### 2.4. Gene Expression Analysis by RT-qPCR

Total RNA was extracted using a method described by Wang et al. [35]. The primers were designed using NCBI Primer BLAST, as described previously [7]. The primer sequences are listed in Appendix A.

### 2.5. Protein Sequence Analysis of GHs

The predicted proteins were analyzed for signal peptides using SignalP v. 5.0 (https://services.healthtech.dtu.dk/service.php?SignalP-5.0, accessed on 24 September 2020) [36]. The presence and location of the GPI-anchoring domains were predicted using PredGPI (http://gpcr.biocomp.unibo.it/predgpi/, accessed on 24 September 2020) [37]. BUSCA was used to predict protein subcellular localization (http://busca.biocomp.unibo.it/, accessed on 24 September 2020) [38].

### 2.6. Statistical Analysis

All experiments were performed in triplicates. The data were analyzed using SPSS 25 software and are expressed as mean ± standard deviation. Trends were considered significant when the mean values of the compared sets differed at *p* < 0.05.

## 3. Results

### 3.1. Polysaccharide Production in G. lucidum under Different Culture Conditions

We selected five time points during culture to investigate the effects of glucose and xylose on the biomass, production of EPS, IPS, and total polysaccharide (TPS) in *G. lucidum*. As shown in Figure 1, glucose is the preferred carbon source for cell growth and polysaccharide production during culture. The maximum production of EPS, IPS, and TPS (41.29 ± 0.67, 116.67 ± 4.84, and 157.96 ± 5.58 mg/g, respectively) was observed when using glucose on the seventh day. The TPS productions using glucose on days 4, 6, and 8 were 2.62, 3.67, and 1.35-fold, respectively, compared with the TPS production using xylose.

### 3.2. DEGs and KEGG Enrichment Analysis

As the different growth conditions were induced by glucose and xylose, 18 cDNA libraries were prepared on days 4, 6, and 8 to reveal the candidate genes and mechanisms underlying polysaccharide biosynthesis. Under a threshold value of |log_2_ fold change| > 1 and *q*-value < 0.05, the number of differentially expressed genes (DEGs) was identified and illustrated. Comparisons of the samples revealed more upregulated DEGs on days 4 and 6 when glucose was used as the carbon source (Figure 2A). Among all the DEGs, 71 DEGs were common under both conditions, whereas 1779 DEGs were upregulated, and 1650 DEGs were downregulated with glucose as the carbon source (Figure 2B). All the DEGs were mapped using the KEGG database. The top 20 enriched pathways are displayed in Appendix A. The amino sugar and nucleotide sugar metabolism pathways were associated with the formation of NDP-sugar precursors in the polysaccharide biosynthetic pathway; however, most enriched genes were associated with chitin biosynthesis.

### 3.3. Gene Expression Dynamics in Nucleotide Sugar Biosynthetic Pathway

Based on the “Amino sugar and nucleotide sugar metabolism” pathway and our previous study, the nucleoside sugar biosynthetic pathway, using glucose and xylose as the carbon source, was proposed in *G. lucidum*. We determined 16 enzymes encoded by 22 genes that synthesize nucleoside sugar precursors in *G. lucidum*. All 22 genes were expressed, and the FPKM (fragments per kilobase of exon model per million mapped fragments) values are listed in Table 1. With glucose as the carbon source, *GL25253-R1* (encoding xylose reductase, XR), *GL29575-R1,* and *GL30389-R1* (both encoding UDP-glucose 4-epimerase) were upregulated on day 6, and *GL26014-R1* (encoding transketolase, TKTA) was upregulated on days 4 and 6. *GL29728-R1* (encoding XR) and *GL31661-R1* (encoding xylulose reductase) were downregulated on day 8.

### 3.4. Expression Dynamics of GT Genes

Based on the reference and gene functional annotations, we identified 80 genes encoding GTs in *G. lucidum*. Among these, 77 genes belonging to 28 families were expressed (FPKM > 1) in this study. The expression dynamics using the FPKM approach and DEGs are shown in Figure 3. Eleven genes showed significantly different expression levels, including seven upregulated and four downregulated genes. Notably, *GL22527-R1*, belonging to the GT20 family and encoding trehalose-6-phosphate phosphatase, was upregulated on all three days with glucose as the carbon source. *GL27844-R1*, belonging to the GT2 family, was more highly expressed with glucose than with xylose; however, its function remains unknown.

### 3.5. Expression Dynamics of GH DEGs

Based on the gene functional annotation, 289 GH genes belonging to 51 families were identified in *G. lucidum*. Among these, 15 were not expressed in this study. Based on this restriction, 98 genes were found to be differentially expressed. The FPKM and log_2_(fold-change) values of all DEGs in the GH family are shown in Figure 4. All DEGs can be classified into six groups according to their functions [39,40]. We determined that 37 genes belonging to 10 GH families were associated with the degradation of fungal and plant cell walls. Moreover, 31 of the 37 genes were upregulated during the last day of glucose treatment. For the fungal cell wall-degrading group, 32 genes showed differential expression, including 21 upregulated genes, 10 downregulated genes, and 1 mixed gene. Additionally, 26 genes were associated with plant cell wall degradation. However, the FPKM values of the 22 genes were low, at 80. The groups whose functions were energy storage and recovery and bacterial cell wall degradation had two genes. In addition, the functions of the two families, including the five genes, were unknown.

### 3.6. Protein Identification and Analysis

Comparative proteomic analyses were performed to identify DEPs between glucose and xylose as the carbon source on day 8. A total of 3040 proteins were identified, including 236 upregulated and 215 downregulated proteins, with a threshold value of |log_2_ (fold change)|>1 and *p* < 0.05. Furthermore, the identified proteins associated with the polysaccharide metabolism were assessed by comparison with the transcriptome on day 8 (Figure 5). In the nucleotide sugar biosynthetic pathway, 20 of the 22 proteins were identified, and 5 were differentially expressed. Among the GT family proteins, twenty-five GTs were detected, including three upregulated proteins (trehalose-6-phosphate phosphatase, ALG9, and glycogen phosphorylase). In addition, 59 proteins belonging to the GH family were detected. Notably, these three proteins were not expressed when glucose was used as the carbon source. Additionally, thirty proteins, including five upregulated and ten downregulated proteins (including one non-expressed protein in glucose), were associated with the degradation of the fungal cell wall as well as the plant cell wall. Seventeen proteins, including five upregulated and two downregulated proteins, were identified in the fungal cell wall-degrading group. Among the 104 identified proteins in the polysaccharide metabolism pathway, 52 were correlated at the transcriptomic level.

### 3.7. Protein Sequence Analysis of the Key GHs

Based on the transcriptomic and proteomic analyses, 28 GH genes and proteins were selected as the significant genes and proteins for polysaccharide biosynthesis (Table 2). The significant genes and proteins were defined by the following standards: (1) the genes or proteins were associated with the degradation of the fungal cell wall or had unknown functions; (2) the genes or proteins were upregulated in the last two days in the transcriptome or upregulated in the proteome. Twenty-nine GH proteins belonged to thirteen GH families. The SignalP 5.1 server analysis revealed that 17 of these proteins contained signal sequences. The GPI anchor was evaluated using the PredGPI prediction server, and five proteins were shown to have potential GPI-anchor sites. The BUSCA program analysis helped predict that eighteen proteins were localized in the extracellular space, five were in the cytoplasm, three were anchored in the plasma membrane, and three were in the plasma membrane.

### 3.8. RT-qPCR Analysis

To validate the transcriptome analysis, we selected six genes to assess the expression dynamics of the two carbon sources using RT-qPCR (Figure 6). In the RT-qPCR analysis, apart from the *PGM* expression on day 4, the transcriptional changes in *PGM* and *UGP* obtained by RT-qPCR were in accordance with those obtained in the transcriptome. GLS (GT48) is also associated with polysaccharide biosynthesis, and the FPKM values of *GL20535-R1* in the transcriptome were increased by 24.98%, 33.62%, and 76.71% in glucose compared to xylose. The relative expression of *GL20535-R1* was increased by 155.49%, 170.05%, and 99.13% in the RT-qPCR results. *GL22527-R1*, *GL27365-R1*, and *GL30087-R1* were upregulated in the transcriptome, and the transcriptional changes were the same as those in the RT-qPCR analysis.

## 4. Discussion

The polysaccharide biosynthetic pathway is complex. Generally, the pathway includes the following steps: the synthesis of NDP-sugar precursors, the assembly of repeating units through GTs, and the polymerization and export of EPS [7,8]. The nucleoside sugar biosynthetic pathway is the most researched and has been constructed in many mushrooms [41,42,43]. Furthermore, the roles of many synthesis-related enzymes have also been elucidated in *G. lucidum*. However, most of the research is focused on the first step of nucleoside sugar biosynthetic pathways to explain the high-yield mechanism of polysaccharide. The further steps, especially the role of GT and GH family genes in the polysaccharide biosynthetic pathway, are not reported in mushrooms. In this study, transcriptomic and proteomic analyses revealed that the key genes that encode PGM, UGP, and GPI were not expressed differently in glucose and xylose cultures. Our results suggested that the genes and proteins in the nucleoside sugar biosynthetic pathway were not highly expressed at either the mRNA or protein levels in the high polysaccharide yield strains. In addition to polysaccharides, nucleoside sugars can also be used to synthesize other compounds such as N-glycans, O-glycans, lipopolysaccharides, saponin, and other glycosides in mushrooms [1,44]. Overall, the role of the genes and proteins in the nucleoside sugar biosynthetic pathway is difficult to investigate for polysaccharide synthesis. 

The biosynthesis of disaccharides, oligosaccharides, and polysaccharides involves the action of different GTs, which catalyze the formation of glycosidic bonds. In this study, *GL22527-R1*, belonging to the GT20 family and encoding trehalose-6-phosphate phosphatase, was determined to be upregulated with glucose as the carbon source, regardless of transcriptomic and proteomic analyses. Trehalose-6-phosphate phosphatase catalyzes the synthesis of trehalose, which is a disaccharide [45]. Chen et al. reported that trehalose-6-phosphate synthase, which can synthesize trehalose in a two-step pathway with trehalose-6-phosphate phosphatase, had a substantial effect on polysaccharide production in *G. lucidum* [46]. In transcriptomic analysis, *GL27844-R1*, belonging to the GT2 family, was highly expressed in glucose. Most cell wall and secreted β-glucans are synthesized by the GT2 family [47]. Hao et al. reported that one GT2 gene (DQGG004795) had higher expression levels in the primordium stage and mature fruiting body stage; this gene belonged to chitin synthase, which may be related to cell wall synthesis [48]. However, the function of *GL27844-R1* must be further elucidated. GLS, belonging to the GT48 family, is an integral membrane protein that is involved in fungal cell wall polysaccharide biosynthesis. High polysaccharide production was related to the upregulation of *GLS* in *G. lucidum* [7,21,49]. Cui et al. reported that the *GFGLS*-silencing transformant of *GFGLS,* iGFGLS-3, decreased mycelial biomass and EPS production (g/L) in *G. frondosa* [23]. In this study, the expression of the GLS gene *GL20535-R1* was increased in both transcriptomic and RT-qPCR analyses when glucose was used as the carbon source. 

GAG, (1→3)-β-glucan, (1→3)-α-glucan, and galactomannan (GM) are well-studied fungal polysaccharides [25,26]. These polysaccharides are found free in the cell wall and external environment, and the synthetic pathways of the EPS are involved in cell wall polysaccharide synthesis [27,28,29]. In both fungal EPS and cell wall polysaccharide synthetic pathways, the GH family of enzymes, such as Sph3 (GH135), Ega3 (GH114), Bgl2p (GH17), Phr1p (GH72), Xog1p (GH5), and Kre6 (GH16), play a vital role [50,51,52]. GHs can hydrolyze polysaccharides as well as branch and elongate the side chains via transglycosylase activities. Hartl et al. reported a GPI-anchor endo-1,3-β-glucanase ENG2 (GH16) in *Aspergillus fumigatus* that has 1,3-β-glucanase and 1,3-β-glucanosyltransferase activities [53]. Few GH79 members also have a GPI-anchor site and modify the fungal cell wall [40]. In the present study, we found several GPI-anchor proteins in the GH16 and GH79 families that could participate in cell wall polysaccharide modification in *G. lucidum*. 

The GH family of enzymes, especially fungal cell wall-degrading enzymes, also participates in several physiological processes in fungi. Sakamoto et al. observed that many cell wall-related enzymes are upregulated after *Lentinula edodes* harvest, such as 1,3-1,6-β-glucan-degrading enzymes in GH5, GH16, GH30, GH55, and GH128 families and chitin-related enzymes in GH18, GH20, and GH75 families [54]. These results suggest that cell wall-degrading enzymes synergistically cooperate for rapid fruiting body autolysis. Zhou et al. found that endo-1,3-β-glucanase (GH16), exo-1,3-β-glucanase (GH2), and 1,3-β-glucosidase (GH55) may act synergistically to completely degrade the (1→3)-β-glucan backbone of the *C. cinerea* cell wall during fruiting body autolysis [55]. Furthermore, β-glucosidase BGL2 (GH3), endo-1,3(4)-β-glucanase ENG16A (GH16), and endo-1,6-β-glucanase (GH30) induce *C. cinerea* stipe cell wall extension [56,57,58,59]. GH152 is a thaumatin-like protein with 1,3-β-glucanase activity and is associated with lentinan degradation and fruiting body senescence [60]. In *G. lucidum*, Wu et al. found glycoside hydrolases could enhance the effect of Tween80 on exopolysaccharide production [49].

In this study, many genes and proteins belonging to the GH3, GH5, GH16, GH17, GH18, GH55, GH79, GH128, GH152, and GH154 families were significantly upregulated in the transcriptome and proteome of *G. lucidum* when glucose was used as the carbon source. These glycoside hydrolases could participate in cell wall extension, senescence, and autolysis by degrading *G. lucidum* cell wall polysaccharides, which was beneficial for extracting more IPS from loose mycelia. Additionally, some of the degraded polysaccharides were released into the culture broth, which was beneficial for obtaining more EPS. These findings provide new insights into the mechanisms underlying the roles of GH family genes in the regulation of high polysaccharide yield in *G. lucidum*.

## Figures and Tables

**Figure 1 microorganisms-11-00772-f001:**
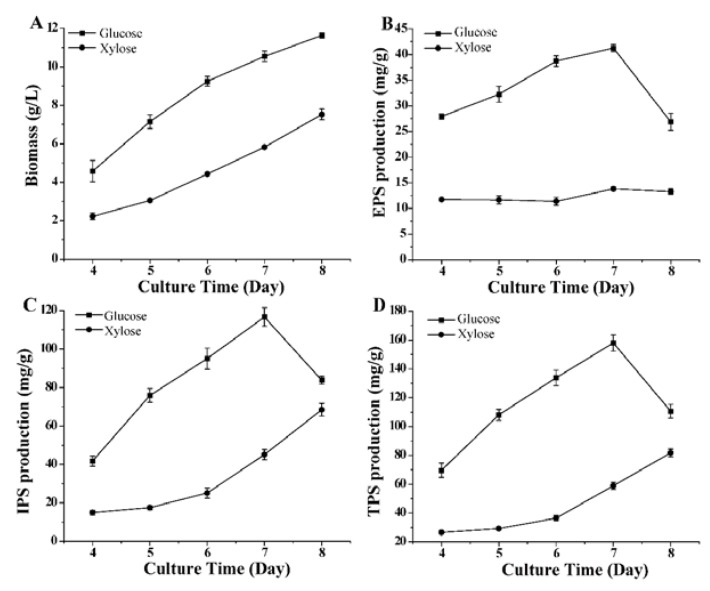
Effect of the carbon sources in culture medium on production of biomass (**A**), extracellular polysaccharides (**B**), intracellular polysaccharides (**C**), and total polysaccharides (**D**) by *Ganoderma lucidum*.

**Figure 2 microorganisms-11-00772-f002:**
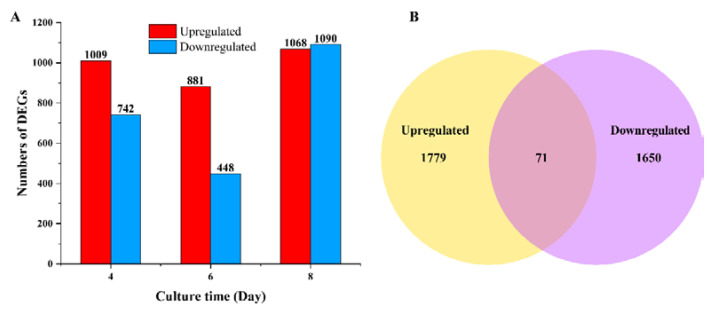
Differentially expressed genes (DEGs) between glucose and xylose as the carbon source. (**A**) The number of DEGs on days 4, 6, and 8. (**B**) Venn diagram of all DEGs for the three days.

**Figure 3 microorganisms-11-00772-f003:**
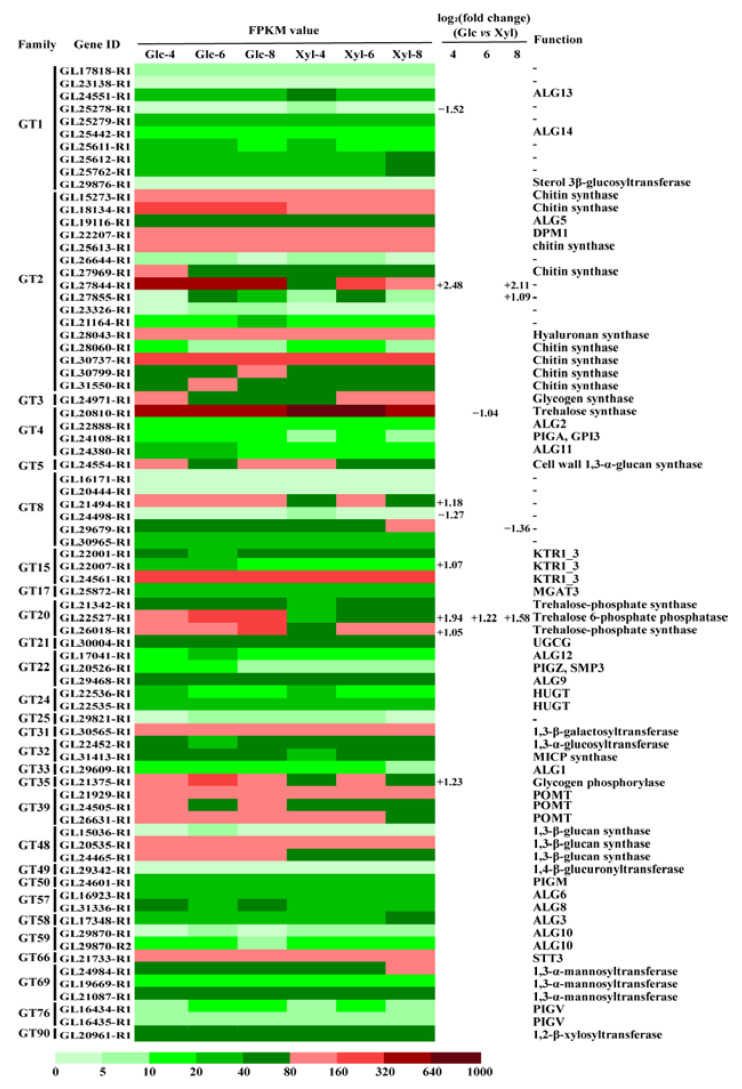
FPKM value, differentially expressed genes, and function of glycosyl transferase family of genes in *G. lucidum* on days 4, 6, and 8. Glc: glucose; Xyl: xylose. +: upregulated; −: downregulated.

**Figure 4 microorganisms-11-00772-f004:**
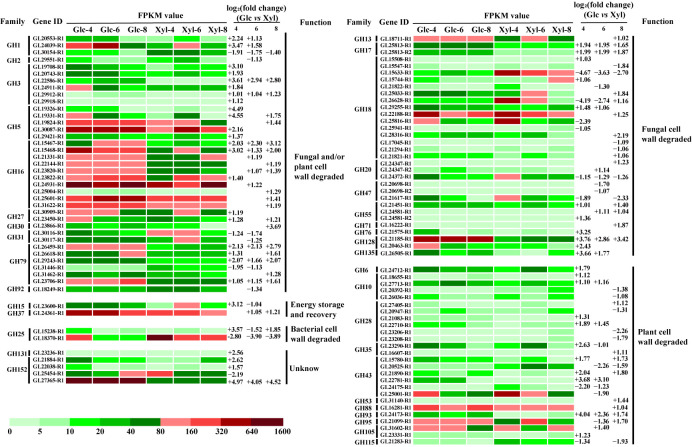
FPKM value and function of the differentially expressed genes in the glycoside hydrolase family of *Ganoderma lucidum* on days 4, 6, and 8. Glc: glucose; Xyl: xylose. +: upregulated; −: downregulated.

**Figure 5 microorganisms-11-00772-f005:**
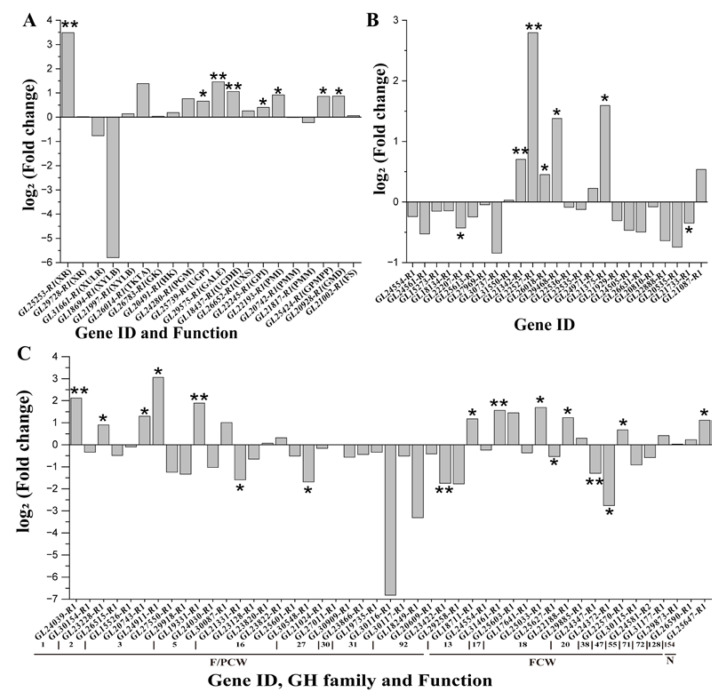
All proteins identified in the polysaccharide biosynthetic pathway of *Ganoderma lucidum* and the correlation analysis of the mRNA and proteins based on the log_2_(fold change). (**A**) The genes and proteins in nucleoside sugar biosynthetic pathway. (**B**) Glycosyltransferases family of genes and proteins. (**C**) Glycoside hydrolase family of genes and proteins. *: Differentially expressed proteins (**: *p* < 0.005; *: 0.005 < *p* < 0.05).

**Figure 6 microorganisms-11-00772-f006:**
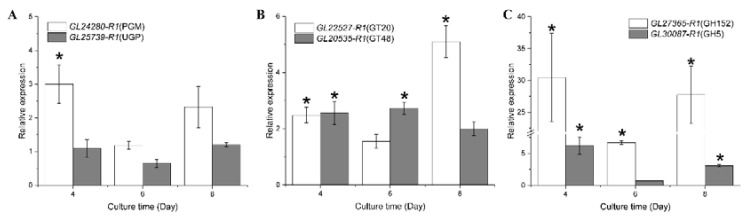
RT-qPCR analysis of genes associated with polysaccharide biosynthesis in *Ganoderma lucidum*. (**A**) The genes in nucleoside sugar biosynthetic pathway. (**B**) Glycosyltransferases family of genes. (**C**) Glycoside hydrolase family of genes. The statistical analysis used for the xylose group was used as a reference (*: |log_2_(Relative expression)| > 1 and *p* < 0.05).

**Table 1 microorganisms-11-00772-t001:** FPKM value of all the genes involved in the nucleotide sugar biosynthetic pathway in *Ganoderma lucidum* during culture.

Gene ID	Gene Name	FPKM Value					
		Glc-4	Glc-6	Glc-8	Xyl-4	Xyl-6	Xyl-8
GL25253-R1	*XR*	161.66	914.25 *	833.84	145.17	43.87	448.15
GL29728-R1	*XR*	284.23	252.43	149.41 *	301.84	321.09	561.03
GL31661-R1	*XULR*	27.35	46.85	24.05 *	30.43	77.50	79.18
GL18694-R1	*XYLB*	94.59	250.12	166.04	174.30	136.04	297.62
GL21997-R1	*XYLB*	101.59	102.22	97.52	129.53	128.80	114.71
GL26014-R1	*TKTA*	62.44 *	52.35 *	54.24	17.52	25.08	32.27
GL26783-R1	*GK*	50.78	51.30	60.06	65.72	61.88	59.12
GL20491-R1	*HK*	26.99	27.72	26.47	29.47	28.72	35.44
GL24280-R1	*PGM*	279.26	265.65	264.92	278.27	308.83	274.51
GL25739-R1	*UGP*	506.58	501.29	416.90	279.62	542.14	348.39
GL29575-R1	*GALE*	75.59	82.80 *	120.30	73.47	23.00	107.93
GL30389-R1	*GALE*	135.67	165.89 *	267.17	180.65	53.07	275.96
GL18437-R1	*UGDH*	257.30	222.58	202.82	272.26	176.89	281.65
GL26652-R1	*UXS*	177.33	174.14	133.19	139.76	144.57	188.09
GL22245-R1	*GPI*	329.20	297.68	245.43	281.02	290.33	386.39
GL22193-R1	*PMI*	139.69	126.56	112.25	93.81	136.62	112.32
GL17878-R1	*PMI*	19.70	13.89	21.62	8.50	19.43	24.38
GL20742-R1	*PMM*	106.29	93.07	77.94	86.58	89.79	93.87
GL21817-R1	*PMM*	98.82	92.67	83.43	88.77	96.24	89.87
GL25424-R1	*GMPP*	133.36	101.89	112.82	114.84	105.41	92.70
GL20928-R1	*GMD*	269.46	251.06	210.68	178.67	214.09	131.50
GL21002-R1	*FS*	91.93	78.47	72.66	80.75	91.68	90.04

XR, xylose reductase; XULR, xylulose reductase; XYLB, xylulokinase; TKTA, transketolase; GK/HK, hexokinase; PGM, α-phosphoglucomutase; UGP, UDP-glucose pyrophosphorylase; GALE, UDP-glucose 4-epimerase; UGDH, UDP-glucose 6-dehydrogenase; UXS, UDP-glucuronate decarboxylase; GPI, phosphoglucose isomerase; PMI, mannose-6-phosphate isomerase; PMM, phosphomannomutase; GMPP, GDP-mannose pyrophosphorylase; GMD, GDP-mannose 4,6-dehydratase; and FS, GDP-fucose synthase. * Differentially expressed genes (DEGs).

**Table 2 microorganisms-11-00772-t002:** Properties of the key glycosyl hydrolase proteins for polysaccharide biosynthesis.

GH Family	Gene ID	Putative Enzymes	Signal Peptide	GPI	Likely Destination
GH1	GL20553-R1	β-glucosidase	No	No	Cytoplasm
	GL24039-R1	β-glucosidase	No	No	Extracellular space
GH3	GL20743-R1	β-glucosidase	No	No	Extracellular space
	GL22586-R1	β-xylosidase	Yes	No	Extracellular space
	GL24911-R1	β-glucosidase	No	No	Extracellular space
	GL29912-R1	β-glucosidase	Yes	No	Extracellular space
GH5	GL19331-R1	-	No	No	Cytoplasm
	GL30087-R1	Exo-1,3-β-glucanase	Yes	No	Extracellular space
GH13	GL18711-R1	Glycogen debranching enzyme	No	No	Cytoplasm
	GL31461-R1	1,4-α-glucan-branching enzyme	No	No	Cytoplasm
GH16	GL15467-R1	Fungal 1,3(4)-β-D-glucanases	Yes	Highly probable	Anchored component of plasma membrane
	GL15468-R1	Fungal 1,3(4)-β-D-glucanases	Yes	Highly probable	Anchored component of plasma membrane
	GL23820-R1	Fungal 1,3(4)-β-D-glucanases	Yes	No	Extracellular space
GH17	GL25813-R1	Exo-1,3-β-glucanase	No	No	Plasma membrane
	GL25813-R2	Exo-1,3-β-glucanase	No	No	Plasma membrane
	GL25603-R1	1,6-β-glucan synthase	Yes	No	Extracellular space
GH18	GL25033-R1	Chitinase	No	No	Extracellular space
	GL29255-R1	Chitinase	Yes	No	Extracellular space
	GL22188-R1	Chitinase	Yes	No	Extracellular space
GH55	GL21451-R1	Exo-1,3-β-glucanase	Yes	No	Extracellular space
	GL24581-R1	Exo-1,3-β-glucanase	Yes	No	Extracellular space
GH79	GL26459-R1	-	Yes	Weakly probable	Extracellular space
	GL26618-R1	-	Yes	Highly probable	Anchored component of plasma membrane
	GL29243-R1	-	Yes	Weakly probable	Plasma membrane
	GL23706-R1	-	No	No	Extracellular space
GH128	GL21185-R1	-	Yes	No	Extracellular space
GH152	GL27365-R1	1,3-β-glucanase	Yes	No	Extracellular space
GH154	GL25647-R1	-	No	No	Cytoplasm

## Data Availability

All raw sequencing data have been deposited in the National Microbiology Data Center (NMDC) under project NMDC10018298.

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
