# Peer review of "Novel Insights into the Mechanism Underlying High Polysaccharide Yield in Submerged Culture of Ganoderma lucidum Revealed by Transcriptome and Proteome Analyses"

_microorganisms, 2023, doi:10.3390/microorganisms11030772_

Round 1

Reviewer 1 Report

The article is devoted to the assessment of the biosynthesis of polysaccharides in the submerged culture of fungus Ganoderma lucidum. There are several points to improve:

1. Summary: "Polysaccharides are important pharmacological components of Ganoderma lucidum" - this is more of a nutraceutical, as they are used mainly in dietary supplements, and not as medicines other than traditional ones.

2. Introduction, line 39: it should be better defined, i.e. "Fungus Ganoderma lucidum (Lingzhi) is a well-known traditional herbal medicine with various medicinal properties"

3. Materials and methods, line 85: The strain G. lucidum CGMCC5.26 was purchased...

4. Lines 108-109: the phenol-sulfuric acid method should e briefly described with the reference to literature.

5. What is known about the polysaccharide composition and purity in the crude extracts? What about the presence of proteins and phenolic compounds and triterpenes? The analytical data and/or references should be added and discussed.

6. Discussion, lines 309-310: GAG (glycosaminoglycans) are polysaccharides from animal connective tissues but not of fungi. Galactomannanns are from legume plants, mushrooms have mannogalactans (MG). The references should be about Ganoderma only not all fungi.

7. All the abbreviations used should be summarised in a separate sections.

Author Response

We appreciative for the detailed feedback that was provided by the reviewers. We have carefully considered each comment and have revised the manuscript point-by-point, accordingly. The revised text is highlighted in yellow throughout the marked (with highlighted) version revised manuscript, for your convenience, and a clear indication of the location of the revision (such as Line..., Page…) is included with each response.

  1. Summary: "Polysaccharides are important pharmacological components of Ganoderma lucidum" - this is more of a nutraceutical, as they are used mainly in dietary supplements, and not as medicines other than traditional ones.

Response: Thank you for your kind suggestion. We have re-written the sentences.

Line 14, Page 1.

  1. Introduction, line 39: it should be better defined, i.e. "Fungus Ganoderma lucidum (Lingzhi) is a well-known traditional herbal medicine with various medicinal properties"

Response: The sentence has been revised.

Line 30, Page 1.

  1. Materials and methods, line 85: The strain G. lucidum CGMCC5.26 was purchased...

Response: We have re-written the sentence.

Line 85, Page 2.

  1. Lines 108-109: the phenol-sulfuric acid method should be briefly described with the reference to literature.

Response: The phenol-sulfuric acid method has been briefly described with the reference to literature.

Line 111-115, Page 3.

  1. What is known about the polysaccharide composition and purity in the crude extracts? What about the presence of proteins and phenolic compounds and triterpenes? The analytical data and/or references should be added and discussed.

Response: Generally, the polysaccharide composition and purity are determined when the polysaccharide need to be separated and purified. In this study, we only determined the content of polysaccharide. The composition and purity of polysaccharide could not affect the polysaccharide production, and the presence of proteins and phenolic compounds and triterpenes could not be detected by the method in this study.

  1. Discussion, lines 309-310: GAG (glycosaminoglycans) are polysaccharides from animal connective tissues but not of fungi. Galactomannanns are from legume plants, mushrooms have mannogalactans (MG). The references should be about Ganoderma only not all fungi.

Response: Thank you for raising this point. GAG (galactosaminogalactan) and galactomannans are the main cell wall polysaccharides and EPSs in some fungi such as Aspergillus fumigatus. However, few studies have focused on the overall pathway about mushroom. In this part, we described other fungi polysaccharides which the biosynthetic pathways are reported, that introduce the current research status.

Line 320-331, Page 11.

  1. All the abbreviations used should be summarised in a separate sections.

Response: According the ‘Instructions for Authors’ of Microorganisms that the following:

Acronyms/Abbreviations/Initialisms should be defined the first time they appear in each of three sections: the abstract; the main text; the first figure or table. When defined for the first time, the acronym/abbreviation/initialism should be added in parentheses after the written-out form.

We have checked all abbreviations based on the ‘Instructions for Authors’, and did not summary the abbreviations in a separated section.’’

Reviewer 2 Report

Dear author,

The study focused on polysaccharide biosynthesis, and the Article was well-designed. On the other hand, the transcriptome and proteome of G. lucidum were significantly regulated. Interestingly, the authors investigated the key genes and proteins associated with high polysaccharide yields in G. lucidum using transcriptome and proteome analyses.

I need some clarifications in your Article entitled “Novel insights into the mechanism underlying high polysaccharide yield in submerged culture of Ganoderma lucidum revealed by transcriptome and proteome analyses,” which are as follows

Abstract

Queries: Line 18: Dear Authors, please note that many eukaryotic phytopathogenic fungi secrete different hydrolytic enzymes to break down the hots cell wall. Fungi secreted various types of cell wall degrading enzymes; most of them are glycoside hydrolases [GHs]. The enzymes that are responsible for breaking down complex carbohydrates and polysaccharides. What is the relationship between cell wall hydrolases and extracellular polysaccharide production from Ganoderma?

What strategies are employed in Ganoderma polysaccharide overproduction/high yield?

Introduction

Lines 39-41 and 59-63: are these lines necessary? Could you explain the biosynthesis of polysaccharides in eukaryotic organisms?

What is the significance of lines 64-74 to the current investigation? Rephrase the sentences.

Please check the previous sentences for a repetition of lines 77-78.

Materials and Methods

Lines 94-97: Exactly rewrite the homogenization and culture processes.

Lines 102-103: Delete the line or rewrite the sentence in accordance with the following sentences. The EPS isolation procedure is not exact; dialysis and other steps may be required.

Discussion 

Lines 283-289. What is the current study's significance [goal and achievement, for example]?

Lines 290-308.

The discussion is unrelated to the current study. Please refer to the research papers pasted below and discuss this research finding.

1. Oehme et al., 2019 

Oehme, D.P.; Shafee, T.; Downton, M.T.; Bacic, A.; Doblin, M.S. Differences in protein structural regions that impact functional specificity in GT2 family β-glucan synthases. PLoS ONE 2019, 14, e0224442 

2. Hao et al., 2022

Hao, H.; Zhang, J.; Wang, Q.; Huang, J.; Juan, J.; Kuai, B.; Feng, Z.; Chen, H. Transcriptome and Differentially Expressed Gene Profiles in Mycelium, Primordium and Fruiting Body Development in Stropharia rugosoannulata. Genes 2022, 13, 1080.

3. Duan et al., 2022

Duan, Y., Han, H., Qi, J. et al. Genome sequencing of Inonotus obliquus reveals insights into candidate genes involved in secondary metabolite biosynthesis. BMC Genomics 23, 314 (2022).

The conclusion did not justify the hypothetical results. Rewrite the sentence to include justification and a title.

"In addition, the results showed that some of the degraded polysaccharides were released into the culture broth, which aided in the production of more EPS."

Author Response

We appreciative for the detailed feedback that was provided by the reviewers. We have carefully considered each comment and have revised the manuscript point-by-point, accordingly. The revised text is highlighted in yellow throughout the marked (with highlighted) version revised manuscript, for your convenience, and a clear indication of the location of the revision (such as Line..., Page…) is included with each response.

  1. Queries: Line 18: Dear Authors, please note that many eukaryotic phytopathogenic fungi secrete different hydrolytic enzymes to break down the hots cell wall. Fungi secreted various types of cell wall degrading enzymes; most of them are glycoside hydrolases [GHs]. The enzymes that are responsible for breaking down complex carbohydrates and polysaccharides. What is the relationship between cell wall hydrolases and extracellular polysaccharide production from Ganoderma?

Response: Thank you for your kind suggestions. The glycoside hydrolases not only break down the hots cell wall but also themselves cell wall. In this study, we found several glycoside hydrolases genes and proteins, which are associated with the degradation of fungal cell wall, were significantly upregulated. These glycoside hydrolases could degrade the self-cell-wall polysaccharide which could be released into the culture broth that increased the extracellular polysaccharide production.

  1. What strategies are employed in Ganoderma polysaccharide overproduction/high yield?

Response: According to the findings in this study, these GHs could be investigated as targets genes, through overexpressing in Ganoderma to improve polysaccharide yield in the future.

  1. Lines 39-41 and 59-63: are these lines necessary? Could you explain the biosynthesis of polysaccharides in eukaryotic organisms?

Response: Line 39-41, Some researchers are proposed that polysaccharide biosynthetic pathway includes these steps in fungi. The sentences have been revised, and the new references have been added.

Line 41-43, Page 1.

Line 59-63, This part has been deleted.

  1. What is the significance of lines 64-74 to the current investigation? Rephrase the sentences.

Response: The polysaccharide biosynthetic pathway is complex which include several steps. However, few studies have focused on the overall pathway about mushroom. In this part, we described other fungi polysaccharides which the biosynthetic pathways are reported, that introduce the current research status. The sentences have been revised.

Line 63-64, Page 2.

  1. Please check the previous sentences for a repetition of lines 77-78.

Response: Thank you for your kind suggestion. We have deleted the sentence.

  1. Lines 94-97: Exactly rewrite the homogenization and culture processes.

Response: The homogenization and culture processes has been re-written.

Line 94-98, Page 2-3.

  1. Lines 102-103: Delete the line or rewrite the sentence in accordance with the following sentences. The EPS isolation procedure is not exact; dialysis and other steps may be required.

Response: In this study, we only determined the content of EPS, not to isolate EPS. And we have briefly described the measurement method in the manuscript.

Line 111-115, Page 3.

  1. Lines 283-289. What is the current study's significance [goal and achievement, for example]?

Response: Thank you for your kind suggestion. The polysaccharide biosynthetic pathway is complex. However, most of the research are focused on the first step of nucleoside sugar biosynthetic pathway to explain the high-yield mechanism of polysaccharide. The further steps, especially the role of GT and GH family genes in polysaccharide biosynthetic pathway, are not reported in mushroom. The current study is mainly investigated the high-yield mechanism of GH in polysaccharide biosynthetic pathway. This part has been re-written, and we hope it given a clear significance.

Line 283-291, Page 10.

  1. Lines 290-308.

The discussion is unrelated to the current study. Please refer to the research papers pasted below and discuss this research finding.

1) Oehme et al., 2019

Oehme, D.P.; Shafee, T.; Downton, M.T.; Bacic, A.; Doblin, M.S. Differences in protein structural regions that impact functional specificity in GT2 family β-glucan synthases. PLoS ONE 2019, 14, e0224442

2) Hao et al., 2022

Hao, H.; Zhang, J.; Wang, Q.; Huang, J.; Juan, J.; Kuai, B.; Feng, Z.; Chen, H. Transcriptome and Differentially Expressed Gene Profiles in Mycelium, Primordium and Fruiting Body Development in Stropharia rugosoannulata. Genes 2022, 13, 1080.

3) Duan et al., 2022

Duan, Y., Han, H., Qi, J. et al. Genome sequencing of Inonotus obliquus reveals insights into candidate genes involved in secondary metabolite biosynthesis. BMC Genomics 23, 314 (2022).

Response: Thank you for the suggestion and references. We have major revised the discussion and added many new references. We have also referred and discussed the three papers.

Line 305-312, Page 10-11.

  1. The conclusion did not justify the hypothetical results. Rewrite the sentence to include justification and a title.

"In addition, the results showed that some of the degraded polysaccharides were released into the culture broth, which aided in the production of more EPS."

Response: The conclusion has been major revised.

Line 348-353, Page 11.

Reviewer 3 Report

1.       English language should be improved thoroughly.

2.       The authors should clearly demonstrate the novelty of this study.

3.       The part of discussion should be much improved, for instance, comparison of polysaccharides derived from other medicinal fungi.

4.       More updated references are required, such as “Si, J.; Meng, G.; Wu, Y.; Ma, H.F.; Cui, B.K.; Dai, Y.C. Medium composition optimization, structural characterization, and antioxidant activity of exopolysaccharides from the medicinal mushroom Ganoderma lingzhi. Int. J. Biol. Macromol. 2019, 124, 1186-1196.” and “Wang, H.; Ma, J.X.; Zhou, M.; Si, J.; Cui, B.K. Current advances and potential trends of the polysaccharides derived from medicinal mushrooms sanghuang. Front. Microbiol. 2022, 13, 965934.”. Please check the references according to the Instructions for Authors.

Author Response

We appreciative for the detailed feedback that was provided by the reviewers. We have carefully considered each comment and have revised the manuscript point-by-point, accordingly. The revised text is highlighted in yellow throughout the marked (with highlighted) version revised manuscript, for your convenience, and a clear indication of the location of the revision (such as Line..., Page…) is included with each response.

  1. English language should be improved thoroughly.

Response: We have sought to improve both the language and academic quality of the manuscript. We have carefully checked and revised the manuscript for grammar and sentence structure, and the language has been polished by experts from a professional language editing service institution.

  1. The authors should clearly demonstrate the novelty of this study.

Response: Thank you for your kind suggestion. The polysaccharide biosynthetic pathway is complex. However, most of the research are focused on the first step of nucleoside sugar biosynthetic pathway to explain the high-yield mechanism of polysaccharide. The further steps, especially the role of GT and GH family genes in polysaccharide biosynthetic pathway, are not reported in mushroom. This study is mainly investigated the high-yield mechanism of GH in polysaccharide biosynthetic pathway, and found many genes and proteins belonging to the GH3, GH5, GH16, GH17, GH18, GH55, GH79, GH128, GH152, and GH154 families were significantly upregulated in the transcriptome and proteome. These findings provide new insights into the mechanisms that GH family genes play an important role to govern high polysaccharide yield in G. lucidum.

We have revised the abstract, discussion and conclusion, and we hope they should clearly demonstrate the novelty of this study.

Line 24-26, Page 1; Line 283-291, Page 10; Line 348-353, Page 11.

  1. The part of discussion should be much improved, for instance, comparison of polysaccharides derived from other medicinal fungi.

Response: Thank you for your kind suggestion. We have major revised the discussion and added many new references from other medicinal fungi.

Line 283-291, Page 10; Line 305-312, Page 10-11;

  1. More updated references are required, such as “Si, J.; Meng, G.; Wu, Y.; Ma, H.F.; Cui, B.K.; Dai, Y.C. Medium composition optimization, structural characterization, and antioxidant activity of exopolysaccharides from the medicinal mushroom Ganoderma lingzhi. Int. J. Biol. Macromol. 2019, 124, 1186-1196.” and “Wang, H.; Ma, J.X.; Zhou, M.; Si, J.; Cui, B.K. Current advances and potential trends of the polysaccharides derived from medicinal mushrooms sanghuang. Front. Microbiol. 2022, 13, 965934.”. Please check the references according to the Instructions for Authors.

Response: Thank you for providing the references. We have updated the references and cited the two papers.

References 6-8, 41-44, 47,48.

Round 2

Reviewer 1 Report

The names of polysaccharides should be unified accodring to the JUPAC rules, for example (13)-β-D-glucan. Similarly, the names of enzymes should be corrected like exo-1,3-β-D-glucanase.

The introduction and discussion should be expanded by more information about polysaccharides of Ganoderma mushrooms using appropriate references.

Author Response

We appreciative for the detailed feedback that was provided by the reviewers. We have carefully considered each comment and have revised the manuscript point-by-point, accordingly. The revised text is highlighted in yellow throughout the marked (with highlighted) version revised manuscript, for your convenience, and a clear indication of the location of the revision (such as Line..., Page…) is included with each response.

  1. The names of polysaccharides should be unified accodring to the JUPAC rules, for example (1→3)-β-D-glucan. Similarly, the names of enzymes should be corrected like exo-1,3-β-D-glucanase.

Response: Thank you for raising this point. All the names of polysaccharides and enzymes have been checked according to the rules. However, some of the names are incomplete. We have tried our best to correct them. For example, it is named α-1,3-glucan based on the reference, then we corrected to (1→3)-α-glucan in the manuscript.

Line 66, Page 2; Figure 3, Page 6; Table 2, Page 9; Line 319, 326, 327, 333, 338, 340, 342, Page 11

  1. The introduction and discussion should be expanded by more information about polysaccharides of Ganoderma mushrooms using appropriate references.

Response: Thank you for your kind suggestion. For further retrieving, we found two new appropriate references and have expanded the introduction and discussion.

Line 49-50, Page 2; Line 313-314, Page 11; Line 343-344, Page 11

Reviewer 3 Report

The authors adequately response and revise the questions for Review 1 point by point. For this reason, I recommend the manuscript could be published in Microorganisms.

Author Response

Thank you for your kind suggestion.